# Association between Lifestyle Behaviours and Mental Health of Adolescents: Evidence from the Canadian HBSC Surveys, 2002–2014

**DOI:** 10.3390/ijerph19116899

**Published:** 2022-06-04

**Authors:** Asaduzzaman Khan, Shanchita R. Khan, Eun-Young Lee

**Affiliations:** 1School of Health and Rehabilitation Sciences, The University of Queensland, Brisbane, QLD 4072, Australia; 2School of Public Health and Social Work, Queensland University of Technology (QUT), Brisbane, QLD 4059, Australia; shanchita.khan@qut.edu.au; 3School of Kinesiology and Health Studies, Queen’s University, Kingston, ON K7L 3N6, Canada; eunyoung.lee@queensu.ca

**Keywords:** mental health, wellbeing, children, sitting time, sedentary behaviour, exercise

## Abstract

Physical activity (PA) and screen time (ST) are associated with mental health in adolescents, though little is known about their inter-relationships. This study examined the associations of PA and ST with psychosomatic complaints in adolescents. Data from four cycles of the Canadian Health Behaviour in School-aged Children (HBSC) surveys, collected between 2002 and 2014, were analysed. Eight psychosomatic health complaints were assessed and dichotomised as frequent (≥2 complaints/week) vs. infrequent. PA was assessed by number of days/week participants were physically active for ≥60 min. Discretionary ST was assessed by adding three screen uses: television, electronic games, and computer. Of the 37,829 adolescents (age 13.74 (SD 1.51) years; 52% girls), 25% boys and 39% girls reported frequent psychosomatic complaints. Multilevel logistic analyses showed that ST levels were positively associated while PA levels were negatively associated with reporting frequent psychosomatic complaints in a dose-dependent manner. Compared with ST ≤ 2 hrs/d, ST ≥ 4 hrs/d in girls and ST ≥ 6 hrs/d in boys showed higher odds of reporting psychosomatic complaints. Participating in PA ≥ 60 min every day compared to no PA showed lower odds of reporting psychosomatic complaints by 44% in girls and 57% in boys. Prospective research is needed to understand the causal pathway of these dose-dependent relationships.

## 1. Introduction

Mental health among adolescents has deteriorated considerably in recent years, especially in many Western countries [1]. In Canada, there was a 9% increase in psychosomatic complaints in adolescents from 8.09% in 2002 to 8.82% in 2018, with a significant year-by-year increase [1]. Analysing data from eight cycles of the annual Canadian Community Health Survey, one recent study reported increases in the prevalence of poor mental health and professionally diagnosed mood and anxiety disorders from 2011 to 2018 [2]. Untreated mental illness can adversely affect a child’s development, educational outcomes, and physical and psychosocial wellbeing [3]. Although the burden of mental health problems spans through the life stages, half of the disorders develop during adolescence by the age of 14 years and 75% by the age of 24 years (Kessler et al., 2005). Given that both lifestyle behaviours and health trajectories are likely to be established during adolescence, identifying modifiable risk factors for mental health during adolescence is critical to informing strategies to aid prevention efforts.

Physical activity (PA) has been associated with various physical and mental health benefits in children and adolescents; regular PA participation is positively associated with self-esteem, quality of life, academic performance, and inversely associated with anxiety, stress, and depressive symptoms in adolescents [4,5]. Despite the benefits of PA, many adolescents are not meeting the recommended levels of PA. For example, a recent report card on physical activity for children and youth in Canada revealed that only 35% of 5–17 year-olds met the recommended 60 min/d of moderate- to vigorous-intensity physical activity (MVPA) [6]. In addition to a lack of PA, children and adolescents are highly engaged with sedentary pursuits that include low-intensity physical activities with low energy expenditure, such as watching television and livestreaming videos or playing video games [7]. For example, 22% of Canadian adolescents aged 12–17 years met the ST recommendation of ≤2 h per day on average [6]. High recreational ST for adolescents has been positively associated with psychosocial difficulties, depressive and anxiety symptoms, poor cognitive performance, attention deficit hyperactivity disorder, social isolation, and sleep difficulties [8,9]. Available research also suggests that a lack of PA engagement and high ST are common in adolescence when levels of PA tend to decrease (in particular among girls) and time spent on screen tends to increase [10]. These lifestyle behaviours are likely to act interactively and/or independently to amplify the risk of poor mental health in school children [8]; however, their dose–response relationship is yet poorly understood. Most of the available research has studied individual behaviour without looking at dose–response relationships. An accurate understanding about the dose–response of relationships is crucial for designing pragmatic strategies to deliver appropriate public health intervention on limiting digital screen use (e.g., whether to limit it at 3 hrs/d or 4 hrs/d) as well as setting a minimum PA level in order to optimise mental wellbeing. Given that lifestyle behaviours have been considered as a low-cost strategy to improve the mental wellbeing of adolescents [11], the current study aimed to examine the associations of PA and ST with psychosomatic complaints of adolescents and whether such relationships were dose-dependent. Gender-stratified analysis was carried out given the gender-based heterogeneity in activity behaviours and mental wellbeing of adolescents [12].

## 2. Materials and Methods

Nationally representative data from four cycles of the Canadian Health Behaviour in School-aged Children (HBSC) surveys, collected in 2002, 2006, 2010, and 2014, were analysed. The HBSC is a collaborative, repeated cross-sectional survey, conducted by WHO, that aims to monitor adolescent health and wellbeing in 50 countries and regions across Europe and North America (for more information, visit http://www.hbsc.org/). The HBSC focuses on understanding young people’s health in their social context—where they live, at school, with family and friends. The HBSC uses a stratified random cluster sampling design and collects school-based data every four years from a nationally representative sample of 11-, 13-, and 15-year-old adolescents in participating countries. Participants provide self-reported data by anonymously completing a questionnaire that includes a range of items on health indicators and related behaviours. Survey administrators in each country received ethics approval from an appropriate regulatory body, and informed consent was obtained from participants and a parent or guardian. Analyses for this manuscript were cleared by The University of Queensland Human Ethics Committee (2021/HE000671).

### 2.1. Outcome Measure

Participants were asked to report how often they experienced eight psychosomatic complaints (i.e., feeling low, irritability or bad temper, feeling nervous, difficulties in falling asleep, feeling dizzy, headache, stomachache, and backache), with five response options: about every day, more than once a week, about every week, about every month, and rarely or never. This instrument for assessing psychosomatic complaints has acceptable test–retest reliability and validity [13]. In this study, the eight items had good internal consistency across survey cycles (Cronbach’s alpha = 82 (80–84)). A summative index was generated and dichotomised as frequent (≥2 complaints weekly) vs. infrequent (≤1 complaint weekly) [14].

### 2.2. Study Factors

Physical activity was assessed by using the number of days the participants were physically active for at least 60 min/day in the last seven days, and the responses were grouped into five categories: none, 1–2 days, 3–4 days, 5–6 days, and 7 days per week. Participants’ ST was assessed using three types of screen-based activities in their free time: (1) watching television, DVDs, other videos (e.g., YouTube); (2) gaming using computers, gaming consoles, tablets (excluding moving or fitness games); and (3) using a computer or other electronic device for other purposes including emailing, social media, chatting, and surfing the internet. ST on weekdays and weekend days were weighted 5:2 to generate an average ST per day, which was then divided into five categories: ≤2 hrs/d, 2–4 hrs/d, 4–6 hrs/d, 6–9 hrs/d, and >9 hrs/d.

### 2.3. Covariates

Covariates were selected based on availability and plausible connection to the outcome measure. Body mass index (BMI; kg/m^2^) was calculated based on self-reported height (cm) and weight (kg). Individual-level socioeconomic status was measured with the Family Affluence Scale (FAS), which is a composite score based on items that assess the households’ number of cars and computers, bedroom sharing, and number of family holidays in the past year. Other covariates included age, gender, alcohol consumption, and survey cycle. 

### 2.4. Statistical Analysis

To examine the associations between PA, ST, and frequent psychosomatic complaints across gender, multilevel modelling was used as the outcome data had a nested structure, with the individual students as the level 1 unit, schools as the level 2 unit, and survey year as the level 3 unit. Specifically, multivariable multilevel logistic regression modelling to examine the association estimates, adjusted for a set of covariates, was used for the overall sample and for boys and girls separately. The model based on the overall sample was additionally adjusted for gender. Finally, we conducted sensitivity analyses using different reference categories of ST (≤4 hrs/d, 4–6 hrs/d, 6–9 hrs/d, and >9 hrs/d) and PA (≤2 days, 3–4 days, 5–6 days, and 7 days per week) to examine whether different categorisations impacted the results. The analysis was conducted using the *runmlwin* command via Stata v17SE (StataCorp, College Station, TX, USA) with binomial logit response models. The association estimates are presented in the form of adjusted odds ratio (aOR) and their 95% confidence interval (CI).

## 3. Results

Descriptive statistics of the analytical sample (*n* = 37,829) are presented in Table 1. The mean age of study participants was 13.7 (SD 1.5) years, and 52% were girls. Overall, 25% boys and 39% girls reported frequent psychosomatic complaints (*p* < 0.001). The percentage of adolescents reporting over 4 hrs/day of ST was 59% for girls and 66% for boys. 

About 18% of the girls reported performing ≥60 min of PA every day compared to 29% of the boys. With progressive increments in ST levels, the prevalence of psychosomatic complaints steadily increased across the four cycles (Figure 1a). Overall, 21.5% of boys and 31.8% of girls with ST ≤ 2 hrs/d reported frequent psychosomatic complaints, while the percentages were 31.2% and 52.2%, respectively, for boys and girls with ST > 9 hrs/d. In contrast, the prevalence of psychosomatic complaints decreased with the increase in PA levels (Figure 1b). Overall, 43.5% of boys and 54.8% of girls with no PA engagement reported frequent psychosomatic complaints, while the percentages were 23.1% and 37.5%, respectively, for boys and girls with PA of ≥60 mins every day.

Multilevel logistic regression modelling, adjusted for the set of covariates, showed that ST levels were positively associated while PA levels were inversely associated with reporting frequent psychosomatic complaints in a dose-dependent manner (Table 2). Compared with ST ≤ 2 hrs/d, the odds of reporting psychosomatic complaints started increasing at ST ≥ 6 hrs/d for boys and ST ≥ 4 hrs/d for girls. For example, odds of frequent psychosomatic complaints in boys were 48% higher (OR 1.48, 95% CI:1.21–1.80) for using ST: 6–9 hrs/d and 85% higher (OR 1.85, 95% CI:1.53–2.25) for using ST > 9 hrs/d when compared with ST ≤ 2 hrs/d. In girls, the respective increases in the odds for frequent psychosomatic complaints were by 59% (aOR 1.59, 95% CI 1.37–1.86) for using ST: 6–9 hrs/d and 126% (aOR 2.26, 95% CI 1.94–2.63) for using ST > 9 hrs/d.

The odds of frequent psychosomatic complaints decreased with the increase in levels of PA for both boys and girls. Compared with no PA engagement, performing more PA (≥60 min/d) decreased the odds of reporting frequent psychosomatic complaints by 53% (aOR 0.47, 95% CI 0.36–0.62) for 3–4 days, 61% (aOR 0.39, 95% CI 0.30–0.51) for 5–6 days, and 57% (aOR 0.43, 95% CI 0.32–0.56) for 7 days per week in boys. In girls, the respective decreases in the odds of frequent psychosomatic complaints were by 49% (aOR 0.51, 95% CI 0.41–0.64) for 3–4 days, 54% (aOR 0.46, 95% CI 0.37–0.57) for 5–6 days, and 44% (aOR 0.56, 95% CI 0.44–0.71) for 7 days per week.

Sensitivity analyses with different reference thresholds used to define the ST and PA categories produced similar results without meaningful changes (Appendix A). Adverse associations of ST with frequent psychosomatic complaints were observed in a dose-dependent manner when compared to ST ≤4 hr/d as a reference for both girls and boys. Although a similar trend was found in the relationship between PA and frequent psychosomatic complaints, the association estimates became slightly lower with the new reference threshold of ≤2 days of PA (≥60 min/d).

## 4. Discussion

This study examined gender-stratified associations of PA and ST with psychosomatic complaints among 37,829 Canadian adolescents who participated in four cycles of the Canadian Health Behaviour in School-aged Children surveys collected between 2002 and 2014. The main findings indicated that, overall, ST levels were positively associated while PA levels were inversely associated with reporting frequent psychosomatic complaints in both boys and girls in a dose-dependent manner. However, dose responsiveness was more consistently observed for ST than PA in both genders. 

There was a significant gender difference in wellbeing, with more girls reporting poor mental wellbeing than boys (39% vs. 25%), as suggested elsewhere [15]. Furthermore, meeting the PA recommendation was higher in boys (29%) than girls (18%), while meeting the ST recommendation was slightly higher in girls (17%) than boys (15%); however, overall, the adherence to movement guidelines was low among Canadian children. Gender-related disparities in PA adherence are well known and observed globally, and Canada is not an exception. The 2020 ParticipACTION report card also indicated that, among those aged 12–17 years, 43% of boys and 17% of girls met the PA recommendation [16]. Combined with generally low adherence to the ST recommendation, promoting PA and reducing ST is a priority for Canadian public health programs and policies, and a particular focus should be placed on health-enhancing behaviours among girls.

Our findings contribute to the limited evidence on dose–response relationships between PA, ST, and poor wellbeing in adolescents, with the association estimates broadly comparable across gender. We found significantly increased odds of reporting frequent psychosomatic complaints among adolescents who reported high ST. A meta-analysis reported a non-linear dose–response association in young people with a higher risk of depression when ST exceeds 2 hrs/d [17]; however, our study showed higher odds of poor wellbeing when ST exceeded 6 hrs/d in boys and 4 hrs/d in girls. Excessive ST has recently been recognised as an emerging risk factor for poorer mental and physical health outcomes among children and youth, independent of PA [18]. 

Our study also showed that an additional 60 min/d of PA was associated with a lower risk of psychosomatic complaints, which is in line with earlier research [11,19,20,21]. Specifically, our study showed that meeting the daily PA recommendations (≥60 min/d) can offer protective effects against psychosomatic complaints for as much as 54% in boys and 41% in girls. Additionally, it appeared that psychosomatic complaints were lowest for adolescents who were engaged in PA ≥ 60 min for 5–6 days per week, and more markedly for girls. “The more PA, the better health outcomes” is a well-known phenomenon in PA promotion, but the benefits of PA may plateau in terms of its potential influence on psychosomatic complaints among Canadian adolescents once they hit a certain level of PA. This finding may be because adolescents who engage in PA on a regular basis already exhibit sufficient levels of PA, thus, showing favourable mental health outcomes.

Based on the data from 42 countries worldwide, low ST and high PA were associated with psychosomatic complaints favourably in dose-responsive and inter-dependent manners [19]. Our findings align well with this large-scale international study. Future research should replicate our findings to build strong evidence of the dose responsiveness of the relationship between PA, ST, and mental health among the Canadian adolescent population. Such an evidence base will inform the future update of Canadian 24-Hour Movement Guidelines and health-promotion policies in Canada. Furthermore, examining the underlying mechanisms through which various PA and ST dimensions (e.g., intensity, duration and type) are linked with the mental health of adolescents is of priority. In a recent large-scale study among 414,489 adolescents from 44 European and North American countries [20], it was found that the inverse and linear association between ST and psychosomatic complaints was stronger for mentally active ST (e.g., playing video games) than the passive ones (e.g., watching TV). A particular emphasis should be on clarifying the differential health impact by gender.

Using nationally representative data from multiple cycles collected between 2002 and 2014, our analyses were controlled for common confounders and examined the simultaneous association of PA and ST with poor wellbeing across gender. However, our results are based on self-reported data and may be subject to biases. Although ST in this study included various screen-based activities, the current analyses did not examine whether ST contexts (e.g., passive vs. mentally active) have different impacts on wellbeing in Canadian adolescents, which deserves further research. The cross-sectional design precludes making any causal inference about the associations, which could be bidirectional. Given the recent rise in sedentary lifestyle, including high screen use, among the Canadian paediatric population [18], the findings highlight the need for prospective studies to understand the causal pathway of the relationships. It is also important to understand the determinants of these lifestyle behaviours to inform interventions.

## 5. Conclusions

This study highlighted that high PA and low ST are associated with fewer psychosomatic complaints among 37,829 Canadian adolescents. Combined with alarmingly low adherence to PA and ST recommendations among Canadian adolescents, promoting health-enhancing lifestyle behaviours should be of public health priority in Canada. Future research should build on our findings by examining different types of PA and ST and their influence on mental health and whether the associations are causal.

## Figures and Tables

**Figure 1 ijerph-19-06899-f001:**
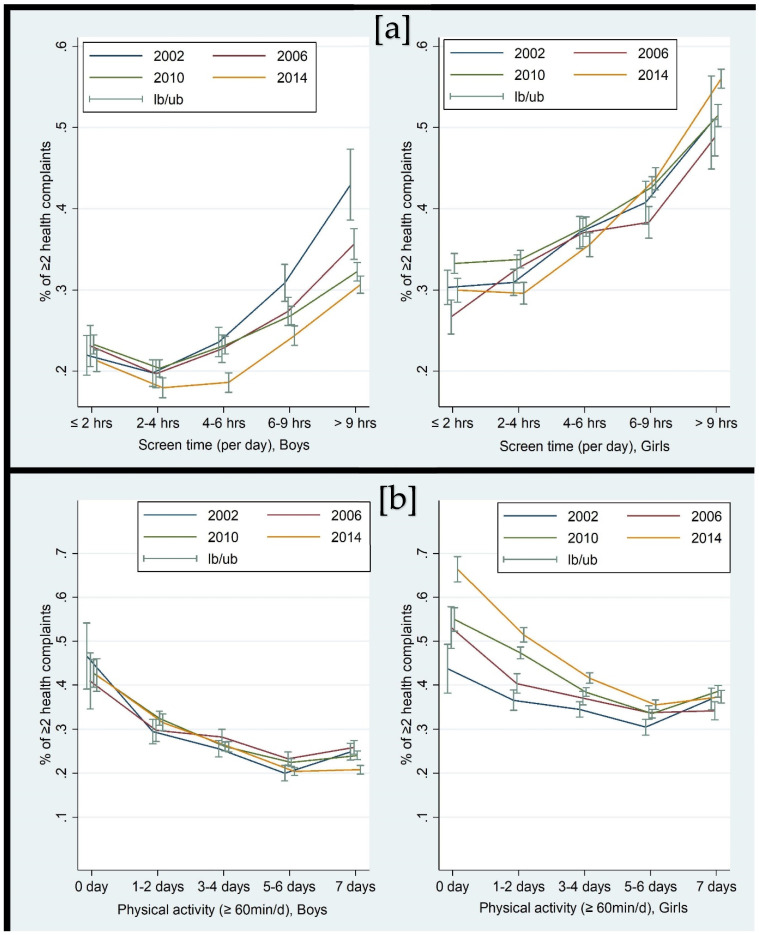
Percent distribution of frequent psychosomatic complaints across different levels of (**a**) screen time and (**b**) physical activity by gender, Canadian HBSC 2002–2014. *Note: lb* represents lower bound; *ub* represents upper bound.

**Table 1 ijerph-19-06899-t001:** Description of study sample, Canadian HBSC study, 2002–2014 (*n* = 37,829).

Characteristics	Boys	Girls
Total participants	18,233	19,596
Mean age (SD)	13.76 (1.51)	13.73 (1.53)
Alcohol intake:		
Never	11,399 (72.54)	11,287 (66.79)
Rarely	2814 (17.91)	3239 (19.17)
Weekly/Monthly	1501 (9.05)	2373 (14.04)
Family affluence scale		
Q1	6098 (36.39)	6918 (37.30)
Q2	3703 (22.09)	4187 (22.57)
Q3	3402 (20.30)	3601 (19.41)
Q4	3557 (21.22)	3842 (20.71)
Mean BMI (SD)	20.92 (4.13)	20.51 (4.01)
Screen time (hrs/d)		
≤2 hrs/d	2771 (15.20)	3351 (17.10)
2–4 hrs/d	3492 (19.15)	4641 (23.68)
4–6 hrs/d	3720 (20.40)	4051 (20.67)
6–9 hrs/d	3959 (21.71)	3791 (19.35)
>9 hrs/d	4291 (23.53)	3762 (19.20)
Physical activity (≥60 min/day)		
0 day	541 (3.01)	806 (4.18)
1–2 days	2084 (11.61)	3260 (16.90)
3–4 days	4441 (24.74)	5843 (30.30)
5–6 days	5653 (31.49)	5970 (30.96)
7 days	5232 (29.15)	3406 (17.66)
Frequent psychosomatic (≥2) complaints	24.99	38.93

SD—standard deviation; Qi—ith quartile. Total may not be equal to 18,233 for boys and 19,596 for girls due to missing data.

**Table 2 ijerph-19-06899-t002:** Adjusted associations between physical activity, screen time, and frequent psychosomatic complaints in Canadian adolescents, HBSC 2002–2014.

Characteristics	Overall	Boys	Girls
	Adjusted OR (95% CI)	Adjusted OR (95% CI)	Adjusted OR (95% CI)
Physical activity (≥60 min/day)			
0 day	1 (Reference)	1 (Reference)	1 (Reference)
1–2 days	0.66 (0.55–0.80)	0.62 (0.47–0.84)	0.69 (0.54–0.87)
3–4 days	0.50 (0.42–0.59)	0.47 (0.36–0.62)	0.51 (0.41–0.64)
5–6 days	0.43 (0.36–0.51)	0.39 (0.30–0.51)	0.46 (0.37–0.57)
7 days	0.49 (0.41–0.59)	0.43 (0.32–0.56)	0.56 (0.44–0.71)
Daily screen time			
≤2 hrs/d	1 (Reference)	1 (Reference)	1 (Reference)
2–4 hrs/d	1.09 (0.97–1.24)	1.09 (0.89–1.34)	1.07 (0.92–1.25)
4–6 hrs/d	1.27 (1.12–1.43)	1.19 (0.97–1.45)	1.30 (1.11–1.51)
6–9 hrs/d	1.57 (1.39–1.77)	1.48 (1.21–1.80)	1.59 (1.37–1.86)
>9 hrs/d	2.08 (1.85–2.35)	1.85 (1.53–2.25)	2.26 (1.94–2.63)

OR—odds ratio; CI—confidence interval. Models were adjusted for age, body mass index, alcohol consumption, family affluence scale (FAS) score, and survey cycle; the model based on the overall sample is additionally adjusted for gender.

## Data Availability

The Health Behaviour in School-aged Children (HBSC) survey data can be accessed on request @ https://www.uib.no/en/hbscdata.

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
