# Peer review of "Association between Lifestyle Behaviours and Mental Health of Adolescents: Evidence from the Canadian HBSC Surveys, 2002–2014"

_ijerph, 2022, doi:10.3390/ijerph19116899_

Round 1
Reviewer 1 Report
Dear authors,
Congratulations for your work.
I appreciate that your paper has an interesting subject and that is well written, following the structure of a good scientific paper.
Regarding the content, I suggest for the Introduction part to explain more about the HBSC studies, in what do they consist, where and how were performed. For this you can read and cite Nemeth A.& collaborators( articles and a book, and the following links:
https://www.jahonline.org/article/S1054-139X(03)00537-8/pdf
For the Discussion part I recommend to take into consideration to refer to more articles and to relate your results to other author' s results. The study to which you compare your result is appropriate but insufficient in my opinion.
I recommend also to rewrite the References following the Author's Instructions of the Journal.
Author Response
Point 1: I suggest for the Introduction part to explain more about the HBSC studies, in what do they consist, where and how were performed. For this you can read and cite Nemeth A.& collaborators( articles and a book, and the following links:
https://www.jahonline.org/article/S1054-139X(03)00537-8/pdf
We thank the reviewer for appreciating our work and offering important comments to improve our presentation.
Response 1: Introduction Based on their comments, we have added the following details in the revised manuscript:
“The HBSC is a collaborative, repeated cross-sectional surveys by WHO that aim to monitor adolescent health and wellbeing in 50 countries and regions across Europe and North America (http://www.hbsc.org/). The HBSC focuses on understanding young people's health in their social context – where they live, at school, with family and friends.” (see Lines 76-79)
Point 2: For the Discussion part I recommend to take into consideration to refer to more articles and to relate your results to other author' s results. The study to which you compare your result is appropriate but insufficient in my opinion.
Response 2: Discussion Following the reviewer comments, we have added an additional reference in the revised manuscript (ref #. 21).
Point 3: I recommend also to rewrite the References following the Author's Instructions of the Journal.
Response 3: References We have revised the references as per the journal styles.
Reviewer 2 Report
This article analyzed the relationship between exercise or screen time habits and mental health in adolescents. The relationship between mental health and exercise or screen time among adolescents is likely to be of interest to readers.
The paper presents most of the necessary information in an uncluttered manner. The argument is consistent throughout, and I don't think it needs any special revision.
In Fig. 1, (a) and (b) are not shown in the figure, and it would be good to show them. Also, please include an explanation of the error bars lb/ub (Lower Bound/Upper Bound). Finally, if possible, since the y-axis is in %, it would be better to use "10, 20..." instead of ".1, .2..." .
Author Response
Point 1:
This article analyzed the relationship between exercise or screen time habits and mental health in adolescents. The relationship between mental health and exercise or screen time among adolescents is likely to be of interest to readers.
The paper presents most of the necessary information in an uncluttered manner. The argument is consistent throughout, and I don't think it needs any special revision.
In Fig. 1, (a) and (b) are not shown in the figure, and it would be good to show them. Also, please include an explanation of the error bars lb/ub (Lower Bound/Upper Bound). Finally, if possible, since the y-axis is in %, it would be better to use "10, 20..." instead of ".1, .2..." .
Response 1: We thank you for appreciating our work and offering useful comments. Accordingly, we have added [a] and [b] to the plot along with a note to explain lb and ub. (see Lines…). We agree that percentages would have been better than decimals in y-axis to represent the data; however, we were unable to find the code to do a quick fix of this presentation. We are happy to invest more time to sort this out if the editor feels this is important.
Reviewer 3 Report
Line 60: The authors state: These lifestyle behaviors are likely to act interactively and/or independently to amplify the risk of poor mental health of school children [8]; however, their dose-response relationship is yet poorly understood. Could the authors please expound further on why this relationship is as of yet poorly understood? It seems like a natural assumption that more screen time will result in increasingly poor outcomes, and that more physical activity results in better health outcomes. Why is it that they say that this relationship is poorly understood?
Line 64: authors state: the current study aimed to examine the associations of PA and ST with psychosomatic complaints of adolescents and whether such relationships were dose-dependent. Can the authors further explain what contribution this makes to the field? How does the question fill a hole in our current understanding of the dangers of screen time and the benefits of physical activity. How does their concept or dose-dependence contribute to a better understanding of what is already known in the field around these variables.
Line 67: The authors state: Gender-stratified analysis was carried 66 out given the gender-based heterogeneity in activity behaviours and mental wellbeing of 67 adolescents [12]. Have the authors considered using the term sex instead of gender given current usage of the term gender. Does their research only refer to sex? Or to gender and sex? Were non-binary sex identifications not considered for this research?
Can the authors explain why they did not consider current medical conditions as covariates?
Author Response
Point 1: Line 60: The authors state: These lifestyle behaviors are likely to act interactively and/or independently to amplify the risk of poor mental health of school children [8]; however, their dose-response relationship is yet poorly understood. Could the authors please expound further on why this relationship is as of yet poorly understood? It seems like a natural assumption that more screen time will result in increasingly poor outcomes, and that more physical activity results in better health outcomes. Why is it that they say that this relationship is poorly understood?
Line 64: authors state: the current study aimed to examine the associations of PA and ST with psychosomatic complaints of adolescents and whether such relationships were dose-dependent. Can the authors further explain what contribution this makes to the field? How does the question fill a hole in our current understanding of the dangers of screen time and the benefits of physical activity. How does their concept or dose-dependence contribute to a better understanding of what is already known in the field around these variables.
Response 1: We thank the reviewer for their thoughtful and constructive comments. Dose-response is an important issue when studying relationships and is also one of the seven Bradford Hill criteria when assessing causality. This information is also crucial for designing public health intervention should the relationship be dose-dependent. For example, most of the current research is assessing screen time with 2 hrs/d recommendation, which is impractical in the current climate when screen use has become an integral part of adolescent life. Knowing the dose-response information helps the program managers and policy makers to design appropriate intervention with accurate dose to optimise mental wellbeing. Please note that we did not make any assumption about the relationship that high screen tine is bad for mental wellbeing, rather we wanted to examine whether the relationship (if any) is dose-dependent. In response, we have added the following in the revised manuscript.
“Most of the available research has studied individual behaviour without looking at dose-response relationships. Accurate understanding about dose-response of relationships is crucial for designing pragmatic strategies to deliver appropriate public health intervention on limiting digital screen use (e.g., whether to limit at 3 hrs/d or 4 hrs/d) as well as setting minimum PA level in order to optimise mental wellbeing.“ (see Lines 63-67)
Point 2: Line 67: The authors state: Gender-stratified analysis was carried 66 out given the gender-based heterogeneity in activity behaviours and mental wellbeing of 67 adolescents [12]. Have the authors considered using the term sex instead of gender given current usage of the term gender. Does their research only refer to sex? Or to gender and sex? Were non-binary sex identifications not considered for this research?
Response 2: Sorry for the confusion caused about gender/sex. We have used gender throughout the manuscript and replaced ‘sex’ by ‘gender’ for consistency.
Point 3: Can the authors explain why they did not consider current medical conditions as covariates?
Response 3: We thank the reviewer for their comment. Unfortunately, the HBSC survey did not collect current medical conditions data and as such we were not able to consider them as covariates in our modeling.
Round 2
Reviewer 3 Report
Authors have addressed comments